# Patient experience and perceived acceptability of whole-body magnetic resonance imaging for staging colorectal and lung cancer compared with current staging scans: a qualitative study

Ruth Evans,[1] Stuart Taylor,[2] Sam Janes,[3] Steve Halligan,[2] Alison Morton,[4] Neal Navani,[3] Alf Oliver,[4] Andrea Rockall,[5,6] Jonathan Teague,[7] Anne Miles,[1] on behalf of the Streamline trials investigators

For numbered affiliations see end of article.

**Correspondence to**
Dr Anne Miles;
ae.miles@bbk.ac.uk

## ABSTRACT

**Objective** To describe the experience and acceptability of whole-body magnetic resonance imaging (WB-MRI) staging compared with standard scans among patients with highly suspected or known colorectal or lung cancer.

**Design** Qualitative study using one-to-one interviews with thematic analysis.

**Setting** Patients recruited from 10 hospitals in London, East and South East England between March 2013 and July 2014.

**Participants** 51 patients (31 male, age range 40–89 years), with varying levels of social deprivation, were recruited consecutively from two parallel clinical trials comparing the diagnostic accuracy and cost-effectiveness of WB-MRI with standard scans for staging colorectal and lung cancer ('Streamline-C' and 'Streamline-L'). WB-MRI was offered as an additional scan as part of the trials.

**Results** In general WB-MRI presented a greater challenge than standard scans, although all but four patients completed the WB-MRI. Key challenges were enclosed space, noise and scan duration; reduced patient tolerance was associated with claustrophobia, pulmonary symptoms and existing comorbidities. Coping strategies facilitated scan tolerance and were grouped into (1) those intended to help with physical and emotional challenges, and (2) those focused on motivation to complete the scan, for example focusing on health benefit. Our study suggests that good staff communication could reduce anxiety and boost coping strategies.

**Conclusions** Although WB-MRI was perceived as more challenging than standard scans, it was sufficiently acceptable and tolerated by most patients to potentially replace them if appropriate.

**Trial registration number** ISRCTN43958015 and ISRCTN50436483.

## Strengths and limitations of this study

► This study is the first to describe the experience and acceptability of having a whole-body MRI (WB-MRI) in patients with highly suspected or known lung or colorectal cancer compared with standard scans.

► Qualitative methodology and the large numbers recruited (with wide age range and deprivation levels) enabled us to observe that there is a range of intensity of difficulties experienced during the WB-MRI.

► Assessment within the context of a research trial, with some patients declining participation due to claustrophobia, might have led to an under-reporting of WB-MRI challenge.

several different imaging tests utilising ionising radiation and potentially adding to the physical and psychological burden of patients with suspected cancer.[1 2] For example it is not unusual for a patient with suspected lung cancer to undergo diagnostic CT, followed by staging positron emission tomography (PET)-CT, dedicated brain imaging and invasive mediastinal nodal sampling prior to the first major treatment decision. More accurate and streamlined staging could improve patient outcomes both by triaging to optimal therapy and decreasing the time between diagnosis and treatment.[3] Whole-body MRI (WB-MRI) has been advocated as a safe, accurate and efficient 'one stop shop' investigation that could potentially replace current complex multimodality staging strategies.[4] A single WB-MRI scan could not only accelerate staging but would simultaneously reduce exposure to ionising radiation, in theory reducing the risk of subsequent radiation-induced malignancies, particularly in

## BACKGROUND

Current National Institute for Health and Care Excellence (NICE) -recommended staging pathways for lung and colorectal cancer are often complex and time-consuming, involving

those diagnosed at a younger age. WB-MRI however can be stressful: its duration, 45–90 min, is longer than standard tests — CT takes a few seconds. MRI scanners are noisy and require full body and head immersion into a relatively narrow tube, often necessitating coils wrapped around the patient that restrict movement. Somewhere between 5% and 30% of patients experience distress associated with the anticipated and actual experience of undergoing MRI.[5–9] Anxiety relates to the scan experience itself as well as the result.[5] Severe claustrophobia can lead to premature scan termination or the need for sedation in 1%–15% of attempts,[10 11] and distress is associated with elevated postscan anxiety,[6] which can engender MRI fear or phobia,[12–14] especially problematic in patients needing future MRI scans.

Increased physical and psychological vulnerability of patients with suspected cancer may reduce their ability to cope with a WB-MRI.[2 15] Cancer diagnosis and subsequent treatment are still much feared despite recognition of better treatment outcomes,[16 17] and patients may be experiencing shock, anxiety and worries about the future.[15 18] While two small qualitative studies have investigated general patient experience of MRI,[19 20] there have been few descriptions of patients undergoing WB-MRI, particularly in those diagnosed with or highly suspected of having cancer.

Patient acceptability is central to the successful adoption of any new technology. Poor acceptability could reduce adherence to WB-MRI, which in turn considerably blunts the impact of the technology, even if diagnostically superior to existing tests.

Two parallel, multicentre, prospective cohort studies[21] investigating the diagnostic accuracy and cost-effectiveness of WB-MRI compared with standard pathways for newly diagnosed lung ('Streamline-L') and colorectal ('Streamline-C') cancer have recently been completed. The aim of this study was to describe patient experience and acceptability of WB-MRI and compare with standard staging tests, in the context of lung and colorectal cancer, using interviews from patients recruited to the Streamline trials. In the trials, the WB-MRI was offered as an additional scan alongside those performed as part of standard care.

## METHODS

There is little existing research describing patients' experiences of undergoing WB-MRI, and we adopted a qualitative design as this provides the flexibility to capture rich descriptions of experiences without a priori knowledge of potential responses. One-to-one interviews were completed face to face (at home or at hospital) or via the telephone. Individual interviews were conducted to facilitate expression of emotions and negative experiences that may be inhibited in a group setting. For two interviews a relative was present to support translation where English was not the patient's first language. Patients were interviewed only once.

### Trial and interview study recruitment

Patients were recruited to the 'Streamline' trials from 10 English National Health Service (NHS) hospitals and consented to undergo WB-MRI in addition to standard staging investigations at one of five MRI scan hospital hubs. During trial consent, patients could opt into the interview study and were contacted by psychology researchers by phone, as soon as possible after all imaging had been completed. Prior to trial consent patients were given information about cancer staging and the WB-MRI; staging was explained to patients as a process where doctors use tests to assess whether cancer has spread around the body to help decide best treatment, and the trial was described as an assessment about whether the WB-MRI may be quicker or better at staging newly diagnosed cancer than standard scans. Accelerated treatment pathways meant some interviews (n=13) were undertaken after treatment had commenced. Trial inclusion/exclusion criteria are published.[21]

The a priori total sample size was a minimum of 50, stratified by cancer site (25 lung and 25 colorectal patients). There was flexibility to recruit beyond this number to achieve saturation, that is, the point beyond which further interviews contribute little new information, although this was not needed. The first 123 trial patients were approached and 91 (74%) agreed to be interviewed. Of the 91 who agreed, 51 patients (56%) participated; we believed we had reached saturation by 50 patients and ceased interviews; however, one additional interview was subsequently completed as a patient (from the original 91 consenting patients) expressed a strong wish to share their views. Reasons patients were not interviewed included consent retracted (n=8), participation in a related questionnaire study (n=14), withdrawal from the main trial (n=12) and interview quota reached before completion of all staging imaging tests (n=6).

### Conducting the interview

Interviews were conducted from March 2013 to July 2014 by two female psychology researchers (RE and AM), independent of the clinical trial team, with prior training and experience of conducting in-depth interviews with patients with cancer.[22 23] The researchers introduced themselves as working with the doctors at the patient's hospital and explained that the interview was an opportunity to provide feedback about the WB-MRI and other tests the patient had had. A topic guide was developed (box 1) to encourage discussion of key aspects of the staging experience, including comparisons between WB-MRI and standard investigations. This guide was reviewed after completion of the first two interviews to assess whether it needed to be modified. Topic ordering and emphasis varied depending on relevance to each patient. Open-ended questions were followed by verbal probing to elicit further clarification. Interviews lasted between 12 and 86 min (mean 48 min), and were recorded digitally. Participating patients were paid £20 plus travel expenses.

| Box 1 Key interview topics |
|---|
| ▶ *I understand that you have been having a number of scans to investigate your symptoms, can you tell me what's been happening?* |
| ▶ *Can you describe what it feels like physically to have the scan?* |
| ▶ *How did you feel during/after/about the scan?* |
| ▶ *What were the staff like?* |
| ▶ *What did you do to cope with [problem identified by patient for example, noise, keeping still, etc]?* |
| ▶ *Would you have another WB-MRI if the doctor recommended it?* |

### Additional measures

Age, gender and postcode data were collated. Postcode was used to calculate an area-based deprivation score using the 2010 Index of Multiple Deprivation (IMD) scale,[24] which was then categorised into quintiles (quintile 1 highest and quintile 5 lowest deprivation).

### Data capture, coding and analysis

Interviews were transcribed verbatim. NVivo V.10 was used to manage data (tagging and labelling) while completing thematic analysis.[25] A coding structure was developed (by RE) using transcript review combined with a reflective log maintained during interviews, and this was reviewed and agreed by both psychology researchers (RE and AM). This framework identified themes related to scan events, beliefs, attitudes and emotional responses, as well as coping strategies. Themes identified were influenced by prior research as well as emergent from the data. This process was iterative, with constant data comparisons to identify similarities and differences within and across individual interviews. Matrix tables were created with themes as columns and relevant data summarised into separate rows from each transcript. Matrix tables provided an overview of all 51 interviews, and ensured that the full range

of experiences were represented in the final description. Patients were not asked to verify the thematic analysis so as not to increase participation burden.

### Statistical analysis

Data were entered into SPSS V.20. Analysis of variance assessed differences in mean ages between lung and colorectal patient groups; $X^2$ assessed group differences in gender, deprivation and interview method; and the Mann-Whitney test was used to assess differences in the median time interval from WB-MRI scan to interview date, between patient groups.

### Ethical review

The trial protocol was reviewed and ethical permission granted by Camden and Islington National Research Ethics Service (NRES) committee on 3rd October 2012 reference numbers: 12/LO/1176 (Streamline-C) and 12/LO/1177 (Streamline-L).

### RESULTS

Table 1 gives full demographics of the interviewed cohort. All but four patients completed the WB-MRI. There were no significant differences in age, gender or area-based deprivation between those who volunteered for interviews (n=91) and those who declined (n=32): mean age: 65 vs 64, $F_{(1,121)}<1$, p=0.527; % male: 64% vs 56%, $X^2$ (df=1, n=123)=0.562, p=0.453; % in deprivation categories (Fisher's exact test $X^2$: 1.834, p=0.788). Similarly there were no significant demographic differences between patients who were (n=51) or were not (n=40) interviewed: mean age 65 vs 64, $F_{(1,89)}<1$, p=0.579; gender % male: 53.4% vs 46.6%, $X^2$ (df=1, n=91)=0.437, p=0.508; % in deprivation categories (Fisher's exact $X^2$: 5.247, p=0.254). The majority of patients were interviewed over

**Table 1** Demographic characteristics of interview participants, by trial stream

| | 'Streamline-L'* % (n=25) | 'Streamline-C'* % (n=26) | Group differences |
|---|---|---|---|
| Gender | | | |
| Male | 60 (15) | 61.5 (16) | $X^2$=0.013, df=1, p=0.910 |
| Female | 40 (10) | 38.5 (10) | |
| Mean age (years) | 65 (SD=11 years) | 64 (SD=9 years) | t=0.261, df=49; p=0.795 |
| Area Deprivation Score (quintile) | | | |
| Highest 1 | 52.0 (13) | 30.8 (8) | $X^2$=5.459 df=2, p=0.065 |
| 2 | 32.0 (8) | 23.1 (6) | |
| Mid to low 3–5 | 16.0 (4) | 46.2 (12) | |
| Interview method | | | |
| Face to face | 20.0 (5) | 26.9 (7) | $X^2$=0.339, df=1; p=0.560 |
| Phone | 80.0 (20) | 73.1 (19) | |
| Median interval between interview and whole-body MRI | 15 days (6–43 days) | 20 days (1–63 days) | U=270.00, n=51, p=0.299 |

*'Streamline-L', lung cancer; 'Streamline-C', colorectal cancer.

the phone (76%, n=39), and the median interval between WB-MRI and interview was 15 days (1–63 days).

### Was the WB-MRI scan a challenge? Diversity of experience and comparison with other scans and tests

As part of the Streamline trials, all patients had WB-MRI as an additional 'trial' investigation alongside standard staging investigations. For patients recruited to 'Streamline-C', standard investigation included CT of the chest, abdomen and pelvis, with some undergoing an additional abdominal/pelvic MRI. Patients recruited to 'Streamline-L' underwent at least a CT and PET-CT, with a proportion undergoing bronchoscopy for tissue sampling.

WB-MRI was perceived to be more challenging than standard scans : *"I would say the most challenging of the MRI scans was the body scan"* s56 (60–69, male, colorectal — 'C'), like *"torture - medieval torture"* s12 (50–59, male, C), while CT was described as *"a walk in the park"* s42 (60–69, male, C) and *"very simple"* s35 (50–59, female, lung — 'L'). PET-CT was described by some as *"so much easier"* s51 (70–79, female, L) and *"wasn't so unpleasant"* s12 (50–59, male, C). However the CT and PET-CT were not without challenges: for example some commented on the intravenous contrast administered during the CT scan, *"you feel a bit, you know, woozy, funny, …and it did make me feel slightly off"* s63 (<50, male, C), and the perceived 'invasiveness' of the radiation exposure associated with CT and PET-CT, *"all that radiation dye in my system and everything…And it just disturbed me. Because I couldn't go near my niece and she's only six"* s54 (<50, female, L). Another patient described the bronchoscopy as more unpleasant: *"I'd rather have them scans again than have that tube down my throat again, that was the worst thing, and I hated it"* s73 (60–69, male, L).

### What was difficult about the WB-MRI?

The factors identified as difficult were claustrophobia, physical discomfort, noise, scan duration, as well as the challenge of coping with emotions elicited during the scan such as fear/panic and isolation. There was a range of reaction intensity, with some reporting positive experiences (table 2).

### Claustrophobia

Patients described feeling trapped or buried in the WB-MRI: *"it was the feeling of being sort of trapped because you can feel the machine all around your body"* s12 (50–59, male, C) and *"I had this sense of being in this sarcophagus"* s17 (60–69, male, L). Only one participant asked to terminate the scan early because of claustrophobia (s17). In comparison the CT scan was described as being like *"going through the big polo mint. And, therefore, it's not as much as like entombment"* s80 (60–69, male, C), and the PET-CT was described as *"not as claustrophobic. That one [MRI] is close to your face. And the other one [PET-CT] is a*

*bit higher. So that, you know, that's not a big problem, really"* s73 (60–69, male, L).

### Noise

Noise emitted from the MRI during data acquisition was aversive for some: *"horrendous. It's horrible!"* s55 (60–69, female, L), but not all, *"it was quite good.… I would say, 'What's coming next? What's the next noise?' So, I enjoyed that"* s18 (60–69, male, C). The absence of noise was a notable feature of other scans, for example, *"PET scan isn't noisy"* s91 (60–69, male, L), and *"there wasn't all the noise. That's the only difference really"* s41 (60–69, male, L).

### Scan duration and physical discomfort

The scan length differentiated WB-MRI from other tests: *"the CT scan is very quick…you're in and out. Whereas, the MRI scan was very long"* s13 (60–69, female, C). However one patient saw the PET-CT scan duration times as equivalent: *"if you took the PET scan, and take the waiting time [for radioactive glucose to circulate], and deduct that, timewise, they're much the same"* s49 (>80, male, L). Duration was linked to comfort: *"the back of my neck was uncomfortable, [the WB-MRI] maybe 15 min too long to be comfortable"* s56 (60–69, male, C). In contrast a patient without discomfort viewed the long duration acceptable: *"I think it's over one and a half hour or something like that, so it was alright for me… is comfortable"* s19 (60–69, male, C).

### Emotions elicited by WB-MRI: WB-MRI induced strong emotions in some

*"I was going to have a sort of panic attack and have to be taken out of the machine…this claustrophobia, the feeling of being… trapped, really"* s12 (50–59, male, C). Worry and fear were also prompted by the novelty of the experience: *"the oppression comes from the unknown"* s45 (70–79, male, C). Dealing with feelings of panic and fear was a challenge in itself and needed effort to overcome: *"when I first laid down in the scan I was frightened…I was trying to put it [fear] out of my head and think positively"* s79 (60–69, female, C). In comparison, *"they [CT and PET-CT] were slightly more easier to relax in that environment, not that I didn't relax in the MRI scan too, it's just that it took a bit more sort of power, more energy if you like, to relax"* s41 (60–69, male, L).

### Cancer context

The WB-MRI was perceived to have the potential to either suggest a cancer diagnosis or reveal additional findings associated with a poorer prognosis and treatment implications: *"I was lying in there and I was thinking, what if they say it's cancer? Then, what?"* s54 (<50, female, L) and *"whether they'd find … it has gone everywhere and you know nothing you can do, I suppose that was the main worry"* s60 (60–69, female, L). One patient had a dilemma whether to go ahead with the scan because of the possibility that its findings might mean he was not offered treatment: *"I was of two minds about the trial scan because I didn't want the trial scan to say the cancer's spread and they wouldn't operate."* s30 (50–59, male, L). However another patient reported relief when the scan confirmed that initial tumours were localised: *"my biggest fear, I know*

**Table 2** Problems and the diversity of challenge experienced related to whole-body MRI

| Problem | No challenge/enjoyable | Low challenge | Medium/high | Did not complete |
|---|---|---|---|---|
| Enclosed space | *"'I'm not claustrophobic.... that (enclosed space) doesn't frighten me"* s36 (50–59, male, L) | *"Maybe in the back of my mind you're worried about it (enclosed space)"* s5 (60–69, female, C) | *"Coffin came to mind...initially, you feel like 'Oh god, it's so close in here'"* s26 (60–69, female, L) | *"I had this sense of being in a sarcophagus"* s17 (60–69, male, L) |
| Noise | *"I thought it like music really... and it relaxed me"* s64 (80+, male, C) | *"the noise was a bit...disconcerting because it was quite loud and quite sort of clanking and banging...and it took some time getting used to it"* s63 (<50, male, C) | *"that's one of the worst ones that I've had to go through with the noise...I felt like something was going to fall off and hit me."* s54(<50, female, L) | |
| Scan duration/discomfort | *"It's actually quite comfortable in there"* s87 (60–69, male, L) | *"It was not excruciatingly painful. It just ached a bit, that is all."* s18 (60–69, male, C) | *"My back was uncomfortable. My, the back of my neck was uncomfortable..." "my brain was saying, 'You can't go much longer...'"* s56 (60–69, male, C) | |
| Negative emotions (eg, anxiety, panic, shock, distress) | *"I mean, there wasn't, you know, anything in particular to be frightening (sic)"* s76 (<50, male, C) | *"when I first went in there, I was quite shocked....I wondered what the hell was going on...after a while I quite enjoyed it"* s18 (60–69, male, C) | *"when I first laid down in the scan I was frightened"* s79 (60–69, female, C), and *"I'm lying there and all of a sudden, they put all these things on top of me... And I'm thinking, 'Oh, my God! ...this is threatening'"* s53 (60–69, male, C) | *"I don't know whether the people doing the scan knew I had lung cancer, I didn't feel any empathy or any sort of friendly attitude. I walked out of there feeling quite upset actually"* s30 (50–59,male, L) |

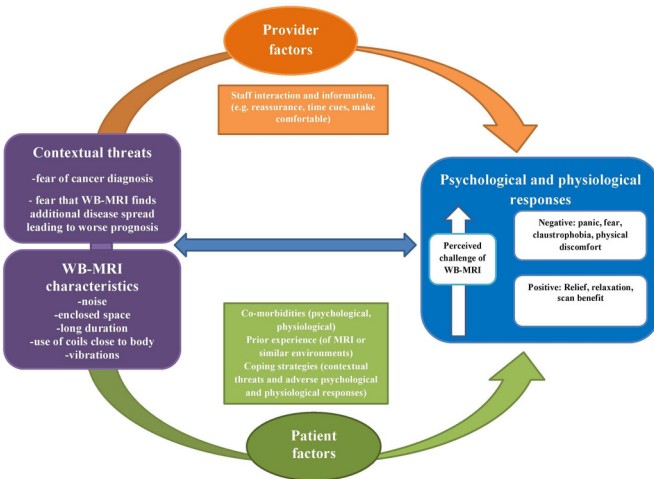

**Figure 1** A model illustrating the factors that influence a patient's whole-body MRI (WB-MRI) experience.

*I've got the cancer in the bowels. I've seen it myself on the colonoscopy. And I was worried that they might, it might just disguise I've got other cancers. So when he said straight away, there was nothing else and I feel relieved"* s45 (70–79, male, C). There was also appreciation of having the WB-MRI because it did reveal spread: "*The MRI actually showed up even more of what's going on with me and I think if I hadn't had it, they would…we still wouldn't know. Because they found traces in five other places I feel a lot better now having the MRI scan, knowing that they've picked up on all of it"* and "*without that, they would…I would just be having the chemo and not the full treatment that I need"* s25 (<50, female, L). The existential threat posed by a cancer diagnosis may have influenced how physical characteristics of the scan were interpreted; one patient described the enclosed experience as "*It's like being in a coffin. You know, I did think that. It's like being in a coffin. But I wasn't. You just have to think that… Because if you've got something really, really bad, it would be like being in a coffin. Just like thinking, 'my God. This is going to be me'. You know. And I would have thought that could be quite scary"* s16 (60–69, male, C); at this point the patient, recently diagnosed with colorectal cancer, became tearful. Another patient interpreted MRI sounds to indicate the presence of pathology: "*And then you'll hear all these noises coming through you, so you wonder…, I thought, if the machine is clicking on, maybe something is wrong there with me"* s13 (60–69, female, C).

### Explaining differences in experiences (see figure 1)
#### Cancer type
Physical symptoms reported more frequently by those recruited to 'Streamline-L' were associated with discomfort — "*I found lying flat on my back, yes, very uncomfortable… I was coping okay for about 20, 25 min and then that's when this pressure or aching or uncomfortable feeling on my chest began to kick in"* s30 (50–59, male, L) — and difficulty with breath holding, "*they had to make me do it a few times because I couldn't hold it for the length of time they wanted me to …*

*I've got COPD, whether that made it worse, I don't know"* s58 (60–69, male, L).

#### Existing musculoskeletal problems
Sometimes these made the WB-MRI more uncomfortable, but not for all: "*I do suffer a bit from back pain… My back started to sort of be pretty uncomfortable"* s42 (60–69, male, C), and "*I was worried about my neck being uncomfortable … because of arthritis but they managed to sort me out and then I managed to last the hour"* s18 (60–69, male, C).

#### Mental health comorbidities
One patient with prior anxiety found the scan difficult, describing their anxiety as causing feelings of claustrophobia: "*I'm not claustrophobic. I just felt that way because of the anxiety. When I get anxious, it feels like everything's just closing in on me"* s54 (<50, female, L).

#### Prior experiences
One patient terminated the scan because he did not like being constrained and his aversion came as a surprise: "*a sense of being constrained and being strapped down which is a new one for me… I haven't encountered that situation before"* s17 (60–69, male, L). In contrast, another participant (s8: 60–69, male, C) who knew he suffered from severe claustrophobia requested a sedative and completed the scan with relative ease. Work experiences meant some patients were used to confined spaces and/or loud noises: "*it doesn't bother me. I've worked in pipes and tunnels and all sorts of places"* s91 (60–69, male, L), and "*if I wasn't in the trade I suppose that that could really freak you out because it's quite loud…"* s76 (<50, male, C). Vicarious experiences were sources of knowledge for patients without prior personal experience, and influenced expectations, sometimes reassuring: "*I have a few friends and relatives… they have already done MRI, so I had something from them, this [is not] painful or anything like that."* s19 (60–69, male, C); sometimes contrary to their own eventual experience, "*it was not as intimidating as I thought… I had heard people talking about going in to an MRI scanner, and they were telling me how nervous they were, and how they hated it… And when I saw it, I thought, 'That's okay'"* s18 (60–69, male, C).

#### Staff contact
Staff supported patients with verbal reassurance, information provision, as well as making them physically comfortable. For example, "*they told me all about it so I didn't get anxious. I knew what to expect"* s26 (60–69, female, L), and "*they managed to find something to put under my neck…they did put it in the right place under my neck and I was okay"* s18 (60–69, male, C). Staff communication acted as a distraction: "*the person speaking keeps my mind occupied"* s26 (60–69, female, L). Some patients spoke of a sense of isolation when in the scanner and hearing a voice confirmed they were not alone "*just made me feel a bit confident that you wasn't on your own, you know?"* s58 (60–69, female, L). Patients varied in whether they felt they received enough information: "*they told me everything I needed to know"* s76 (<50, male, C) versus "*there wasn't any [information]…that*

*was the lacking part"* s35 (50–59, female, L). Whether staff were perceived to be reassuring was also important for some and a change in staff midway through the scan was detrimental: "*she was constantly talking to me, I was fine. But then … another lady took over and she didn't talk as much. … when I panicked*" s58 (60–69, female, L), and "*People I've been speaking with beforehand… I've actually built up the initial trust… Then, suddenly, a complete stranger is now telling me to breathe in and out… then said, 'I should be coming down now to inject you with dye'… In my mind, I thought the absolutely bizarre. You know, he killed everyone in the ward. Now he's out to kill the patients.*" s80 (60–69, male, C). Failure to heed a patient preference influenced their decision to terminate the scan: "*…they told me that they were going to inject dye. And once again I said 'don't put that cannula in my left arm' but she did and the vein collapsed. And I said to them 'I can't do any more now… I've had enough'*" s30 (50–59, male, L).

### Coping

Patients adopted various coping strategies, categorised into two main groups: (1) strategies for coping with distressing thoughts, emotional responses or physical sensations, for example, mental distraction and relaxation — "*Because I love Cyprus I was thinking of Cyprus the whole time. And it sort of takes your mind off of what's going*" s55 (60–69, female, L), and "*breathing, breathing…And you know trying to sort of just to calm……Because you can't help but get anxious*" s60 (60–69, female, L) — and (2) strategies related to motivation to complete the scan, such as focusing on benefit beliefs: "*I was so fed up with pain that I would have done…you know, any investigation was better… it was [WB-MRI] something to get me better*" s14 (60–69, female, C), "*I was laying there and I was thinking, 'Well, it's for my own benefit so I can live with that'*" s8 (60–69, male, C), and "*knowing I had cancer, you know, that's probably the most frightening thing that probably anyone could be told… So, this [WB-MRI] was all a culmination of things that's going to help me get better….The bed rocking a little bit, these loud noises really paled into insignificance because in my body now, I've got a nasty little house guest, which has now stayed, not welcome. I'm going to rid. And this is part of the mechanism to get rid…I was totally focused on the cancer. And these are the pictures that would help me get that done*" s80 (60–69, male, C).

Patients undergoing the WB-MRI scan were not only faced with the challenge of a potentially new technology, but were also coping with a possible or recent cancer diagnosis. For some, this influenced the challenge posed by the WB-MRI: "*it's been you know really anxious time. It's not just going to have an MRI. You obviously know something is wrong with and you're hoping that there's only going to be one thing wrong with you and then you get two things [tests prior to WB-MRI found a spine tumour in addition to lung cancer]. And all of a sudden, it's a completely different ballgame you know for me. About that stage, I had realised that I probably wasn't going to be able have the operation and just carry on with my life as normal. … which is why I was finding it so hard to relax. It wasn't the MRI, it was just me. …you're stressed out*" s60 (60–69, female, L).

### Willingness to have another WB-MRI scan

All patients stated they would undergo another WB-MRI scan if the doctor recommended it, including the four patients who had requested scan termination. However this agreement was offered with varying enthusiasm: "*well if I have to…I don't like it one bit but if it has to be done*" s73 (60–69, male, L), compared with "*yes without hesitation*" s63 (<50, male, C).

## CONCLUSIONS

We aimed to describe patient experience and acceptability of WB-MRI as a potential replacement for the modalities currently used for staging lung and colorectal cancer. Patients recognised that WB-MRI was different from other scans, although the extent of the challenge varied considerably. The majority completed WB-MRI with just four patients requesting premature termination; all were prepared to attempt future WB-MRI. Patients adopted a variety of coping strategies, and these centred on the physical and emotional responses experienced during scanning, as well as focusing on beliefs that bolstered motivation to complete the scan.

To our knowledge, there are only two other comparable qualitative studies of patients' experiences of MRI.[19 20] In common with our study the enclosed space, noise, physical discomfort and duration were identified as challenging, and the need for staff support was also highlighted. The importance of a 'trustful dialogue' between radiographer and patients, to facilitate coping strategies,[19] resonates with comments made by patients in our study.

Noise and confinement are the most frequently cited negative aspects of MRI in quantitative work published over the last 20 years.[6 12] Other studies identify additional factors increasing the likelihood of premature scan termination, including acquisition position (head first, and/or prone); age and gender (middle age or female); prior experience (first scan or prior negative experience); and comorbidity (pain, anxiety).[6 13 26 27] Our results concur with some of these findings; we found some evidence that comorbidity (eg, anxiety and physical symptoms) increased scan challenge, although there were contrary examples where musculoskeletal patients did not experience anticipated discomfort after adjustments were made by staff. Unlike others studies (eg, refs [12 14]) prior experience (positive or negative) generally increased coping ability.

Our study adds to the existing literature. Our patient cohort was highly suspected or just diagnosed with cancer, and we were able to elicit influences of this on their experience of MRI. Furthermore the specific scan type under investigation (WB-MRI) differs from simpler MRI scans such as knees and spine, based on its longer duration and need for receiver coils to cover the whole body. Patients described how the WB-MRI had the potential to reveal a cancer diagnosis or additional metastatic disease, with treatment implications. This caused anxiety and a dilemma as to whether to have

the scan. However some patients also described relief and gratitude for having the scan either when it found no additional cancer or indeed when it diagnosed additional metastatic disease. The perceived length of the scan, together with its title of 'whole body', may have added to patients perceptions about its ability to influence their treatment and prognosis. The WB-MRI scanner was likened to a coffin or sarcophagus, and it is possible that these observations were more emotive because the patients were contemplating a life-threatening diagnosis. Carlsson and Carlsson[19] reported greater fear when MRI was used to confirm serious illness. WB-MRI for cancer staging comes at a point when patients are likely to be emotionally vulnerable, stretching coping resources for challenging medical procedures. However, while the implications of scanning patients with comorbidities and emotional vulnerability should be acknowledged, it is important to note that most completed the WB-MRI.

Our study suggests that good staff communication could reduce anxiety and boost coping strategies by acting as a source of distraction, motivation and emotional reassurance. The varied experiences encountered underline the need for staff interaction to be tailored. Advanced staff training to build rapport can reduce MRI non-completion rates and increase patient satisfaction.[28 29] A recent UK survey suggests that while strategies are in place to help reduce MRI-related anxiety (commonly written leaflets, verbal prescan information, music during scanning, dedicated staff support), these are not optimised yet, with resource restrictions a potential barrier to implementation.[30]

## Study limitations

The generalisability of our results may be limited. WB-MRI was performed within the context of a research trial and some patients declined participation citing claustrophobia. Acceptability of WB-MRI may be lower than our study suggests. However the patients were from a wide age range and deprivation levels, and were interviewed from a number of different hospitals. As a result, a diverse spread of experiences were documented and included some who ultimately were unable to tolerate scanning.

## Clinical implications

Our study confirms that WB-MRI can be a challenging experience and that staff support is important in modifying scan-related stress. Our study also highlights that patient experience is varied, and for the most part the scan was tolerated, suggesting WB-MRI could potentially replace existing modalities, assuming adequate diagnostic accuracy and cost-effectiveness. Experience of WB-MRI reflects the success of coping strategies adopted by patients and the quality of support they receive. Further research would facilitate understanding of the interrelationship between patients' experiences, the effectiveness of their coping strategies and of the support received from staff, such that patients and clinicians can benefit maximally from the diagnostic potential of WB-MRI.

**Author affiliations**
[1]Department of Psychological Sciences, Birkbeck University of London, London, United Kingdom
[2]Division of Medicine, Centre for Medical Imaging, University College London, London, United Kingdom
[3]Division of Medicine, Lungs for Living Research Centre, University College London, London, United Kingdom
[4]C/O National Cancer Research Institute, London, United Kingdom
[5]Department of Surgery and Cancer, Imperial College, London, United Kingdom
[6]Department of Radiology, Royal Marsden NHS Foundation Trust, London, United Kingdom
[7]Cancer Research UK & UCL Clinical Trials Centre, London, United Kingdom

**Acknowledgements** The authors would like to acknowledge the contribution of the Cancer Research UK and UCL Clinical Trials Centre.

**Collaborators** The authors of this paper are part of a wider group that form the Streamline trials investigators and include the following collaborators: A Aboagye, L Agoramoorthy, S Ahmed, A Amadi, G Anand, G Atkin, A Austria, S Ball, F Bazari, R Beable, S Beare, H Beedham, T Beeston, N Bharwani, G Bhatnagar, A Bhowmik, L Blakeway, D Blunt, P Boavida, D Boisfer, D Breen, J Bridgewater, S Burke, R Butawan, Y Campbell, E Chang, D Chao, S Chukundah, B Collins, C Collins, V Conteh, J Couture, J Crosbie, H Curtis, A Daniel, L Davis, K Desai, M Duggan, S Ellis, C Elton, A Engledow, C Everitt, S Ferdous, A Frow, M Furneaux, N Gibbons, R Glynne-Jones, A Gogbashian, V Goh, S Gourtsoyianni, A Green, Laura Green, Liz Green, A Groves, A Guthrie, E Hadley, A Hameeduddin, G Hanid, S Hans, B Hans, A Higginson, L Honeyfield, H Hughes, J Hughes, L Hurl, E Isaac, M Jackson, A Jalloh, R Jannapureddy, A Jayme, A Johnson, E Johnson, P Julka, J Kalasthry, E Karapanagiotou, S Karp, C Kay, J Kellaway, S Khan, D Koh, T Light, P Limbu, S Lock, I Locke, T Loke, A Lowe, N Lucas, S Maheswaran, S Mallett, E Marwood, J McGowan, F Mckirdy, T Mills-Baldock, T Moon, V Morgan, S Morris, S Nasseri, P Nichols, C Norman, E Ntala, A Nunes, A Obichere, J O'Donohue, I Olaleye, A Onajobi, T O'Shaughnessy, A Padhani, H Pardoe, W Partridge, U Patel, K Perry, W Piga, D Prezzi, K Prior, S Punwani, J Pyers, H Rafiee, F Rahman, I Rajanpandian, S Ramesh, S Raouf, K Reczko, A Reinhardt, D Robinson, P Russell, K Sargus, E Scurr, K Shahabuddin, A Sharp, B Shepherd, K Shiu, H Sidhu, I Simcock, C Simeon, A Smith, D Smith, D Snell, J Spence, R Srirajaskanthan, V Stachini, S Stegner, J Stirling, N Strickland, K Tarver, M Thaha, M Train, S Tulmuntaha, N Tunariu, K van Ree, A Verjee, C Wanstall, S Weir, S Wijeyekoon, J Wilson, S Wilson, T Win, L Woodrow, D Yu.

**Contributors** ST, SJ, SH, AMM, NN, AO, AR and AM designed the clinical trial protocol, which includes this study. JT assisted in patient recruitment. RE and AM interviewed patients, analysed and interpreted the data. RE wrote the first draft of the manuscript. All authors reviewed the manuscript and approved the final draft.

**Funding** This project was funded by the National Institute of Health Research health technology assessment NIHR HTA programme (project number 10/68/01) and will be published in full in Health Technology Assessment. The project is supported by researchers at the National Institute for Health Research University College London Hospitals Biomedical Research Centre. S.Taylor and S Halligan are NIHR senior investigators. SM Janes is a Wellcome Trust Senior Fellow in Clinical Science.

**Disclaimer** This report presents independent research commissioned by the National Institute for Health Research (NIHR). The views and opinions expressed by authors in this publication are those of the authors and do not necessarily reflect those of the NHS, the NIHR, NETSCC or the HTA programme or the Department of Health. The views and opinions expressed by the interviewees in this publication are those of the interviewees and do not necessarily reflect those of the authors, those of the NHS, the NIHR, MRC, CCF, NETSCC or the HTA programme or the Department of Health.

**Competing interests** None declared.

**Ethics approval** NRES Committee London — Camden and Islington.

**Provenance and peer review** Not commissioned; externally peer reviewed.

**Data sharing statement** Additional data from anonymised transcripts are available by emailing ae.miles@bbk.ac.uk.

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
