## [Reviewer comments · BMJ Open]

ARTICLE DETAILS

TITLE (PROVISIONAL)	Patient experience and perceived acceptability of whole body magnetic resonance imaging for staging colorectal and lung cancer compared with current staging scans: a qualitative study
AUTHORS	Evans, Ruth; Taylor, Stuart; Janes, Sam; Halligan, Steve; Morton, Alison; Navani, Neal; Oliver, Alf; Rockall, Andrea; Teague, Jonathan; Miles, Anne

VERSION 1 - REVIEW

REVIEWER	Jon Banks University of Bristol, UK
REVIEW RETURNED	06-Mar-2017

GENERAL COMMENTS	On one level this paper is fine, the methods are okay bar a few minor issues that I have highlighted below but I am concerned whether it is giving substantive insight into the research question. I am not sure that I can answer that but I will lay out my concerns below, 1. The abstract – generally okay but the statement about staff interaction in the results is not supported by the results data in the main paper.2. Background – the authors have not set out in sufficient detail what the current staging procedures are, what a WB-MRI offers in comparison with existing practice and what this could mean for patients. In other words, what's in it for patients and why should they take this test rather than the others? I appreciate that this is probably the point of the main trial but it would still be helpful to provide more context on potential benefits.3. In the final para of background the authors link acceptability with adherence which feels out of place here in relation to a diagnostic investigation (arguably adherence could relate to completing the MRI but this should be explicit)4. In methods the authors state that some interviews took place after treatment had started. I think they should give details of how many and they should also give details of the range of time duration overall from MRI to interview.5. The recruitment numbers are not clear – I don't understand how you got from 123 to 91 to 51 nor do I understand where the extra one came from that you describe6. Page 11 – was the MRI a challenge. The first para is not clear – I presume all those who are listed as xx-c and xx-l had an WB-MRI
--

	but it does not read clearly. 7. Page 17 – whether staff were perceived to be reassuring was also important. This is okay but the emphasis given to the staff role in the abstract and conclusion is not supported by this rather thin section in the results. 8. My biggest issue with the paper is I'm not sure that the findings are sufficiently tied into the experience of cancer. Whilst the authors argue that the disease heightens and accentuates fears around MRI (for some) but I don't feel this is sufficiently developed. The problem is that in reading these results I feel that it could have been reporting the perspectives of a sample of the general population on WB-MRI which may well have been covered in previous research
--	---

REVIEWER	Sara Reis Teixeira Hospital das Clinicas, Ribeirao Preto Medical School, University of Sao Paulo - Brazil
REVIEW RETURNED	02-Apr-2017

GENERAL COMMENTS	Thanks for sending me the manuscript to review. Acceptability of a method for staging cancer by the patients is an important issue to be addressed when trying to replace a method for other. Overall, the study was well conducted, clearly and objectively written, though there is an issue regarding the aim of the study. The objectives written in the abstract "To determine the acceptability of whole body magnetic resonance imaging (WB-MRI) staging among patients with highly suspected or known colorectal or lung cancer" (page 4 line 5) do not match the objectives written in the manuscript "The aim of this study was to compare patient experience and acceptability of WB-MRI with standard staging tests, in the context of lung and colon cancer using patient interviews" (page 7 line 18). If you ought to compare acceptability and experience of WB-MRI to the standard methods used, it would be more appropriate to include paragraphs/sections showing this comparison. I did not find any mention to the comparison to other methods. For example, there was no section comparing WB-MRI with bronchoscopy or PET, which I think might not be well tolerated by some patients for the same reasons, such as time consuming. This comparison would enhance the interesting in reading and importance of your work.
---

VERSION 1 – AUTHOR RESPONSE

Reviewer 1:

1. "The abstract – generally okay but the statement about staff interaction in the results is not supported by the results data in the main paper".

We agree with this comment and so we have now added additional verbatim quotes in the results (page18: line 27, to page 19: line 14) to support our assertion that staff interaction influences how challenging patients find the whole-body MRI scan (WB-MRI).

2. "Background – the authors have not set out in sufficient detail what the current staging procedures are, what a WB-MRI offers in comparison with existing practice and what this could mean for patients. In other words, what's in it for patients and why should they take this test rather than the others? I appreciate that this is probably the point of the main trial but it would still be helpful to provide more

context on potential benefits”.

We have added additional text to the introduction section (page 5: line 7 to 16) to better explain the potential benefits to patients of WB-MRI. In particular we provide an example of the current staging procedures for lung cancer as an example of the existing complexity of staging procedures patients’ experience. In addition we have also added text outlining the potential advantages of the WB-MRI to the patient in comparison with existing staging scans (page 5: line 25 to 31).

3. “In the final para of background the authors link acceptability with adherence which feels out of place here in relation to a diagnostic investigation (arguably adherence could relate to completing the MRI but this should be explicit).”

We have additional text to make clear the link is between WB-MRI acceptability and WB-MRI adherence only (page 6: line 21 to 22).

4. “In methods the authors state that some interviews took place after treatment had started. I think they should give details of how many and they should also give details of the range of time duration overall from MRI to interview”.

We have amended the manuscript to give details of the number of interviews that took place after treatment had commenced (page 7, line 18) and the median time interval between MRI and interview, for all participants combined (page 9, line 49 to 50).

5. “The recruitment numbers are not clear – I don’t understand how you got from 123 to 91 to 51 nor do I understand where the extra one came from that you describe.”

We have modified the text in the methods to improve clarity (page 7: 33 to 49) about recruitment numbers. We now only list the numbers of patients and reasons for not being interviewed, for those who initially agreed to participate (n=91) when first approached at the trial recruitment stage. We have given further details about the circumstances around why the ‘51st’ participant was interviewed (page 7: line 38 to 42).

6. “Page 11 – was the MRI a challenge. The first para is not clear – I presume all those who are listed as xx-c and xx-l had an WB-MRI but it does not read clearly”.

We have added additional text to clarify that all patients had WB-MRI as an additional ‘trial’ investigation alongside standard staging investigations” (page 11: 9 to 12).

7. “Page 17 – whether staff were perceived to be reassuring was also important. This is okay but the emphasis given to the staff role in the abstract and conclusion is not supported by this rather thin section in the results”.

In the results section we have now given extra details as described above in relation to point 1.

8. “My biggest issue with the paper is I’m not sure that the findings are sufficiently tied into the experience of cancer. Whilst the authors argue that the disease heightens and accentuates fears around MRI (for some) but I don’t feel this is sufficiently developed. The problem is that in reading these results I feel that it could have been reporting the perspectives of a sample of the general population on WB-MRI which may well have been covered in previous research.”

We agree with the reviewer that we should further develop the concept of the influence of a cancer diagnosis in our manuscript. In response, we have included a new results sub-section called ‘Cancer Context’ (page 15: line 52 to page 16: line 55) where we have included additional verbatim quotes and developed themes that highlight how being investigated for, or recently diagnosed with, cancer can increase the challenge of the WB-MRI for some. Again, this was a strong theme throughout many of the interviews and we agree was unrepresented in the original manuscript. In addition, in the Conclusions, we better explain how our findings are novel (in particular see page 21: line 37 to page 22 up to line 18). In part this is because of the specific –homogenous- cancer patient group studied (unlike more heterogeneous previous qualitative work), and in part because we are studying a novel technology-whole body MRI. Most previous work has dealt with standard MRI, typically musculoskeletal or neurology, which are single organ scans usually of short duration. Conversely WB-MRI is a prolonged scan (often 1 hour) requiring whole body submersion with receiver coils. It

therefore represents a greater challenge than standard MRI (line/page), which is captured by our results. We have now included a diagram which depicts the different influences, we found, on patient experience of WB-MRI including the contextual threats.

Reviewer 2:

1. "Overall, the study was well conducted, clearly and objectively written, though there is an issue regarding the aim of the study. The objectives written in the abstract "To determine the acceptability of whole body magnetic resonance imaging (WB-MRI) staging among patients with highly suspected or known colorectal or lung cancer" (page 4 line 5) do not match the objectives written in the manuscript "The aim of this study was to compare patient experience and acceptability of WB-MRI with standard staging tests, in the context of lung and colon cancer using patient interviews (page 7 line 18)"

Thank you for this observation. We have now altered the text so that the objective in the abstract also specifies the comparison of the WB-MRI with standard staging tests (see page 1: line 5 to 9), which was part of our aim.

2. "If you ought to compare acceptability and experience of WB-MRI to the standard methods used, it would be more appropriate to include paragraphs/sections showing this comparison. I did not find any mention to the comparison to other methods. For example, there was no section comparing WB-MRI with bronchoscopy or PET, which I think might not be well tolerated by some patients for the same reasons, such as time consuming. This comparison would enhance the interesting in reading and importance of your work."

We agree with this observation and have now added additional text and verbatim quotes, in the results sections (see in particular pages 11 to 15) related to comparisons made by patients between the challenges of WB- MRI and those associated with standard scans and procedures. As noted by the reviewer, we now better highlight that the standard staging scans and tests were not without their own challenges. In the interviews patients were encouraged to discuss their experiences of both the WB-MRI and the standard staging scans, and our updated results we feel better reflect this now.

VERSION 2 – REVIEW

REVIEWER	Jon Banks University of Bristol, UK
REVIEW RETURNED	22-May-2017

GENERAL COMMENTS	I've had another look at the paper and am happy with most of the changes made by the authors which have markedly improved the readability of the paper and its relevant to the field. I have some minor points which I feel need attention prior to publication but I do not need to see the paper again. 1. On Page 12 the authors have made it clear that the WB-MRI is undertaken by patients as an additional trial investigation. This is very clear and a very important point which I feel should be much more upfront in the paper (specifically at the end of the background section where the rationale and research focus is explained) and in the abstract. 2. In the methods there is a section on statistical analysis with no explanation of what the listed tests are used for, it's fairly clear in the results section but I think it should be explicit 3. On page 16 where the authors describe the significance of the cancer context they describe patients perceiving the test to have as having the potential to give a poorer prognosis. I think a bit more context is needed here, patients will have participated in the trial on the basis that WB-MRI offers the potential for something. It's
---

	important here to understand whether the information they got from the trial literature is influencing this perspective 4. On page 21, the line that 'positive interactions bolstered coping while negative interactions had the opposite effect' borders on tautology. The paragraph goes on to discuss how a 'trustful dialogue' helps patients to cope with some of the challenges associated with WB-MRI that have been identified - this is a more nuanced approach and where the focus should be. 5. I am not convinced by the reasoning around co-morbidity and prior experience influencing experience - that line does not seem to concur with the section in the results which suggest more ambivalent results. 6. There is a lot of repetition in the conclusion, particularly around staff interaction and support.
--	---

REVIEWER	Sara Reis Teixeira Consultant Paediatric Radiologist Clinical Hospital, Ribeirao Preto Medical School, University of Sao Paulo Brazil
REVIEW RETURNED	18-May-2017

GENERAL COMMENTS	Thanks for the opportunity to review this manuscript. The authors replied to the reviewer's comments and included amendments which improved the manuscript. Every issue was solved with clear sentences and paragraphs.
---

VERSION 2 – AUTHOR RESPONSE

Responses to suggestions for change offered by reviewer 1 (please note page reference relates to the marked copy):

1. "On Page 12 the authors have made it clear that the WB-MRI is undertaken by patients as an additional trial investigation. This is very clear and a very important point which I feel should be much more upfront in the paper (specifically at the end of the background section where the rationale and research focus is explained) and in the abstract". We have now added a sentence to both the abstract (page 4) and the background section (page 7) which states that WB-MRI was offered as an additional scan as part of the trials.

2. "In the methods there is a section on statistical analysis with no explanation of what the listed tests are used for, it's fairly clear in the results section but I think it should be explicit" We have now added an explanation of the statistical tests used (page 10/11).

3. "On page 16 where the authors describe the significance of the cancer context they describe patients perceiving the test to have as having the potential to give a poorer prognosis. I think a bit more context is needed here, patients will have participated in the trial on the basis that WB-MRI offers the potential for something. It's important here to understand whether the information they got from the trial literature is influencing this perspective." We have now added a description of the information patients were given about cancer staging and the WB-MRI trial, prior to consent to the trial. Staging was explained to patients as a process where doctors use tests to assess whether cancer has spread around the body to help decide best treatment, the trial was described as an

assessment about whether the WB-MRI may be quicker or better at staging newly diagnosed cancer than standard (page 8, methods section).

4. "On page 21, the line that 'positive interactions bolstered coping while negative interactions had the opposite effect' borders on tautology. The paragraph goes on to discuss how a 'trustful dialogue' helps patients to cope with some of the challenges associated with WB-MRI that have been identified - this is a more nuanced approach and where the focus should be." We have deleted the sentence quoted above and replaced with text in both the Abstract and Conclusion which focuses on how staff can help patients through the challenge of the WB-MRI (see page 4 and 24).

5. "I am not convinced by the reasoning around co-morbidity and prior experience influencing experience - that line does not seem to concur with the section in the results which suggest more ambivalent results." We have added additional text to the Conclusion to reflect the variability in results relating to co-morbidity and prior experience (page 23).

6. "There is a lot of repetition in the conclusion, particularly around staff interaction and support." We have reviewed the Conclusion section and we are keen to keep most of the current content, however we have edited to reduce the number of references related to the impact of staff interaction (see pages 22-24).